# Thermodynamic Analysis of Air-Cycle Refrigeration Systems with Expansion Work Recovery for Compartment Air Conditioning

Xiaoling Yang, Zhefeng Wang, Ze Zhang, Shuangtao Chen, Yu Hou and Liang Chen *

State Key Laboratory of Multiphase Flow in Power Engineering, Xi'an Jiaotong University, Xi'an 710049, China; yxl1375055173@stu.xjtu.edu.cn (X.Y.); w1z2f3@stu.xjtu.edu.cn (Z.W.); zhangze@xjtu.edu.cn (Z.Z.); stchen.xjtu@xjtu.edu.cn (S.C.); yuhou@mail.xjtu.edu.cn (Y.H.)
* Correspondence: liangchen@mail.xjtu.edu.cn; Tel.: +86-029-8266-4921

**Abstract:** As a requirement for sustainable development, air-cycle refrigeration has received wide attention as a candidate for environmentally friendly air conditioning technology. In this study, the thermodynamic performance of air refrigeration cycles is investigated in compartment air conditioning. The effects of compressor efficiency, expander efficiency, ambient humidity, all-fresh-air supply and ambient pressure on the cycle performance are presented. The effects of compressor arrangement in the high-pressure cycle and the low-pressure cycle are compared. An open-loop high-pressure cycle has a larger COP than that of an open-loop low-pressure cycle but requires larger heat exchange. The performance of air refrigeration cycles with full fresh air is studied, and the influence of fresh air is discussed. Schemes for condensed water recirculation with wet compression are proposed, which can improve the COPs of open-loop low-pressure cycles by 44.7%, 48.8% and 48.4%. In the air conditioning of plateau trains, open-loop high-pressure cycles have slightly lower COPs, but they can supply air with elevated pressure and oxygen concentration.

**Keywords:** air cycle; reverse Brayton refrigerator; fresh air conditioner; COP



## 1. Introduction

The reverse Brayton refrigerator has been employed in space refrigeration [1,2], aircraft environmental control systems [3,4], superconducting [5,6], comparative power generation systems [7] and cryogenic refrigeration. Due to the increasing attention to environmental issues, air-cycle refrigerators based on the reverse-Brayton refrigeration cycle have received widespread attention as one of the most promising long-term solutions to achieve sustainable refrigeration [8–14].

In recent decades, one of the most pressing concerns for air conditioning has been indoor air quality, particularly with the emergence of epidemics such as the SARS virus and the novel coronavirus (SARS-CoV-2), and the fresh-air conditioner has been paid wide attention as an effective means to prevent virus accumulation in rooms. In traditional fresh-air systems, there is a return-air heat exchanger or mixer, which may pollute the fresh air. In order to improve air quality and avoid cross-infection, the all-fresh-air system has become necessary for many public buildings. This dedicated outdoor air system (DOAS), as a fresh-air system with no return air and low supplied temperature, has received much attention since its emergence [15–17]. At present, the DOAS has been used in public buildings such as hospitals and schools. The air-cycle refrigeration system offers the direct treatment of outdoor air, including the elimination of harmful substances such as viruses and pollutants, which makes it a promising candidate for a DOAS.

In high-altitude trains, the ambient pressure and density of ambient air are low, especially when the train travels from the plain to the plateau, and the variation in ambient pressure makes people uncomfortable. The problem is solved by maintaining higher indoor

air pressure. High supplied-pressure is required to maintain indoor pressure, which is almost impossible to achieve in traditional air conditioning systems. In contrast, the air refrigeration system can achieve high supplied-pressure and recover the pressure energy of indoor exhaust air. In addition, the high-pressure air in an air-refrigeration cycle can be used to generate high-concentration oxygen.

In a reverse Brayton refrigerator with a co-axial configuration of the turbo-expander and compressor, the expansion work can be directly recovered into the compression process. Both Li [18] and Catalano [9] proposed a scheme to integrate a compressor and an expander in order to recover the expansion work and improve the cycle performance. Hou, Li and Zhang [10] studied open-loop air-cycle refrigeration using seawater cooling on ships and found that the efficiency of the compressor and expander has a critical influence on the system COP. Hou and Zhang [11] further studied a low-temperature open-loop reverse Brayton-cycle air refrigerator with circulating water cooling and analyzed the impact of the compressor efficiency, expander efficiency and expander inlet temperature on the system COP. With circulating water cooling, the cycle had a higher COP. However, these two studies can only be applied to situations where a water source is available and cannot be directly applied to general, compartment air conditioning. Parker [19] established a calculation model for the open-loop reverse Brayton-cycle air refrigerator by assuming the efficiency of components and studying the system performance under variable conditions.

Most of the previous studies of reverse Brayton-cycle refrigerators are mainly focused on cryogenic applications [20–26]. Although some studies [10,11,27] considered the effect of humidity, there is no comparative study of wet-air refrigeration cycles with different configurations. The all-fresh-air air-conditioner based on the reverse Brayton cycle has not been studied yet. The characteristics of the air refrigeration cycle under low ambient pressure have not been studied either. This study aimed to investigate the characteristics of the reverse Brayton-cycle air refrigerator for room temperature cooling. The influence of the humidity on the system COP is analyzed, and a scheme to improve the COP of the system by recycling condensed water is further proposed. The characteristics of the air refrigeration cycle utilized in all-fresh-air systems and low ambient pressure are studied.

## 2. Theoretical Analysis of Air Refrigeration Cycles

Air refrigeration cycles can be configured with three types of reverse Brayton cycle, including the closed cycle, the open-loop low-pressure cycle (TC cycle) and the open-loop high-pressure cycle (CT cycle), as shown in Figure 1. In the open-loop low-pressure cycle, air of ambient temperature can directly expand to low-temperature air of sub-ambient pressure, and the hot-side heat exchanger is not used. After the cooling power is consumed in a cold-side heat exchanger, the air is compressed and discharged into the outside. In the open-loop high-pressure cycle, the air after expansion is directly supplied to the room, and the cold-side heat exchanger is not used. The condensed water can be removed in the hot-side heat exchanger or after the expander in the case of high-humidity ambient air, and low-temperature, dry air is supplied to the room.

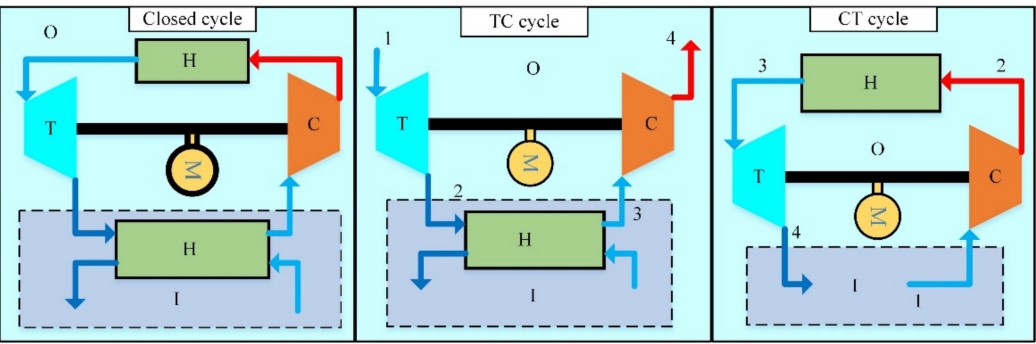

**Figure 1.** Schematic of air refrigeration cycles. T: turbo-expander; C: compressor; H: heat exchanger; I: indoor; O: outdoor.



The coefficient of performance (COP) of the cycle is defined as the ratio of the cooling capacity to the power consumption of the system. In order to simplify the calculation, the pressure drop in the pipes and the heat exchanger is ignored. The compression ratio and expansion ratio (*Er*) in the cycle are the same, and the working fluid is air which is considered an ideal gas.

In the closed cycle, the output shaft power $W_1$ and input shaft power of the cycle $W_2$ are calculated as Equations (1) and (2), respectively.

$$W_T = q_m \frac{k}{k-1} R T_{AC} \left[ 1 - (Er)^{\frac{1-k}{k}} \right] \times \eta_T \tag{1}$$

$$W_C = q_m \frac{k}{k-1} R T_{CI} \left[ (Er)^{\frac{k-1}{k}} - 1 \right] / \eta_C \tag{2}$$

The cooling capacity of cycle is determined by Equation (3):

$$Q_c = q_m (h_{CI} - h_{EO}) = q_m (h_{CI} - h_{AC} + h_{AC} - h_{EO}) = q_m \left[ h_{CI} - h_{AC} + \frac{k}{k-1} R T_{AC} \left( 1 - Er^{\frac{1-k}{k}} \right) \times \eta_T \right] \tag{3}$$

The COP of the cycle is calculated by Equation (4):

$$COP_{Closedcycle} = \eta_C \frac{(T_{CI} - T_{AC}) + \eta_T T_{AC} \left( 1 - Er^{\frac{1-k}{k}} \right)}{\left[ T_{CI} Er^{\frac{k-1}{k}} - T_{CI} - \eta_C \eta_T \eta_M T_{AC} \left( 1 - Er^{\frac{1-k}{k}} \right) \right]} = \eta_C \frac{\eta_T T_{AC} \left( Er^{\frac{k-1}{k}} - 1 \right) - (T_{AC} - T_{CI}) Er^{\frac{k-1}{k}}}{\left( T_{CI} Er^{\frac{k-1}{k}} - \eta_T \eta_C \eta_M T_{AC} \right) \left( Er^{\frac{k-1}{k}} - 1 \right)} \tag{4}$$

Similarly, the COP of the open-loop low-pressure cycle and high-pressure cycle is determined by Equations (5) and (6):

$$COP_{TCcycle} = \eta_C \frac{(T_{CI} - T_O) + \eta_T T_O \left( 1 - Er^{\frac{1-k}{k}} \right)}{\left[ T_{CI} Er^{\frac{k-1}{k}} - T_{CI} - \eta_C \eta_T \eta_M T_O \left( 1 - Er^{\frac{1-k}{k}} \right) \right]} = \eta_C \frac{\eta_T T_O \left( Er^{\frac{k-1}{k}} - 1 \right) - Er^{\frac{k-1}{k}} (T_O - T_{CI})}{\left( T_{CI} Er^{\frac{k-1}{k}} - \eta_C \eta_T \eta_M T_O \right) \left( Er^{\frac{k-1}{k}} - 1 \right)} \tag{5}$$

$$COP_{CTcycle} = \eta_C \frac{(T_I - T_{AC}) + \eta_T T_{AC} \left( 1 - Er^{\frac{1-k}{k}} \right)}{\left[ T_I Er^{\frac{k-1}{k}} - T_I - \eta_C \eta_T \eta_M T_{AC} \left( 1 - Er^{\frac{1-k}{k}} \right) \right]} = \eta_C \frac{\eta_T T_{AC} \left( Er^{\frac{k-1}{k}} - 1 \right) - (T_{AC} - T_I) Er^{\frac{k-1}{k}}}{\left( T_I Er^{\frac{k-1}{k}} - \eta_C \eta_T \eta_M T_{AC} \right) \left( Er^{\frac{k-1}{k}} - 1 \right)} \tag{6}$$

where $T_{AC}$ is the air temperature after the hot-side heat exchanger, and $T_{CI}$ is the air temperature after the cold-side heat exchanger. $T_I$ is the indoor temperature, and $T_O$ is the outdoor temperature.

It is clear that the mechanical efficiency $\eta_M$, compressor isentropic efficiency $\eta_C$ and turbo-expander isentropic efficiency $\eta_T$ determine the system performance. Moreover, the expander efficiency has a relatively greater impact, which not only affects the output work of the expander, but also determines the enthalpy drop in the expander and hence the cooling capacity of the system. Another factor affecting system performance is the expansion ratio *Er*, or the compression ratio. As the expansion ratio increases, the enthalpy drop in the expander increases, leading to an increase in the cooling capacity of the system. However, the raised expansion ratio also increases the power consumption, and the value should be optimized to obtain the maximum COP of the cycle.

By taking the partial derivative of COP with respect to $Er^{(k-1)/k}$, the optimal *Er* can be obtained by making the derivative equal to zero. It can be found that the optimal *Er* satisfies Equation (7):

$$Er^{\frac{k-1}{k}}_{Closedcycle,\,opt} = \frac{\eta_T \frac{T_{AC}}{T_{CI}} + \sqrt{\left( \eta_T \frac{T_{AC}}{T_{CI}} \right)^2 - \left( 1 - \frac{T_{AC}}{T_{CI}} + \eta_T \frac{T_{AC}}{T_{CI}} \right) \left[ \eta_T \frac{T_{AC}}{T_{CI}} + \eta_C \eta_T \eta_M \frac{T_{AC}}{T_{CI}} \left( \frac{T_{AC}}{T_{CI}} - 1 \right) \right]}}{1 - \frac{T_{AC}}{T_{CI}} + \eta_T \frac{T_{AC}}{T_{CI}}} \tag{7}$$

Similarly, the optimal expansion ratios of the open-loop low-pressure and high-pressure cycles can be obtained by Equations (8) and (9):

$$Er_{\text{TCcycle, opt}} = \left[\frac{\eta_T \frac{T_O}{T_{CI}} + \sqrt{\left(\eta_T \frac{T_O}{T_{CI}}\right)^2 - \left(1 - \frac{T_O}{T_{CI}} + \eta_T \frac{T_O}{T_{CI}}\right)\left[\eta_T \frac{T_O}{T_{CI}} + \eta_C \eta_T \eta_M \frac{T_O}{T_{CI}}\left(\frac{T_O}{T_{CI}} - 1\right)\right]}}{1 - \frac{T_O}{T_{CI}} + \eta_T \frac{T_O}{T_{CI}}}\right]^{\frac{k}{k-1}} \tag{8}$$

$$Er_{\text{CTcycle, opt}} = \left[\frac{\eta_T \frac{T_{AC}}{T_I} + \sqrt{\left(\eta_T \frac{T_{AC}}{T_I}\right)^2 - \left(1 - \frac{T_{AC}}{T_I} + \eta_T \frac{T_{AC}}{T_I}\right)\left[\eta_T \frac{T_{AC}}{T_I} + \eta_C \eta_T \eta_M \frac{T_{AC}}{T_I}\left(\frac{T_{AC}}{T_I} - 1\right)\right]}}{1 - \frac{T_{AC}}{T_I} + \eta_T \frac{T_{AC}}{T_I}}\right]^{\frac{k}{k-1}} \tag{9}$$

When $T_I$ = 300.15 K, $T_O$ = 308.15 K, $T_{AC}$ = 313.15 K, $T_{CI}$ = 295.15 K, $\eta_T$ = 0.8, $\eta_C$ = 0.75, $\eta_M$ = 0.99, it can be obtained that $Er_{\text{Closed cycle, opt}}$ = 2.27, $COP_{\text{Closed cycle, max}}$ = 0.659. For the open-loop low-pressure cycle, it can be obtained that $Er_{\text{TC cycle, opt}}$ = 1.96, $COP_{\text{TC cycle, max}}$ = 0.739. For the open-loop high-pressure cycle it can be obtained that $Er_{\text{CT cycle, opt}}$ = 1.94, $COP_{\text{CT cycle, max}}$ = 0.743. Due to the temperature difference in the heat exchanger, $T_{CI}$ is lower than the indoor temperature $T_I$, and $T_{AC}$ is higher than the outdoor temperature $T_O$, leading to a lower COP of the closed cycle than that of the open-loop cycles.

As the discharge pressure and temperature for one-stage compression is relatively high, a two-stage compression with intercooling can be employed to reduce the compression power. By integrating the expander with one of the compressors, the matching characteristics of the aerodynamic performance can be improved while the expansion work is recovered. Therefore, the TC cycle can be modified to the TC-C cycle and C-TC cycle, and the CT cycle can be modified to the C-CT cycle and CT-C cycle, as shown in Figure 2.

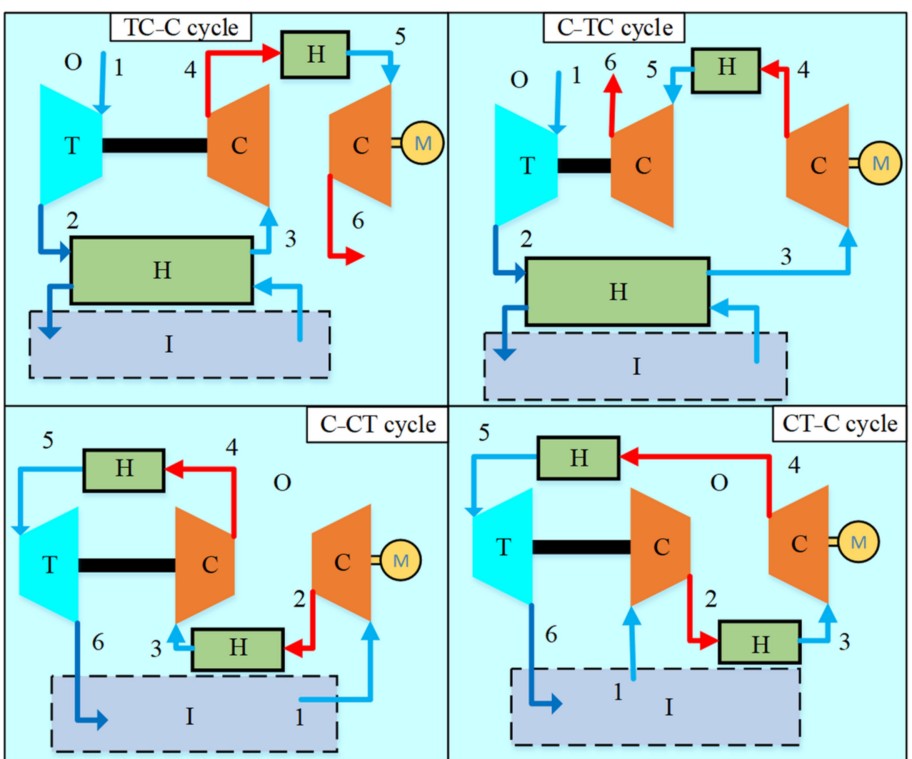

**Figure 2.** Schematic of open-loop air refrigeration cycles with two-stage compressor.

The COP of the two open-loop low-pressure refrigeration cycles (TC-C cycle and C-TC cycle) can be determined by Equations (10) and (11), respectively:

$$\text{COP}_{\text{TC-Ccycle}} = \eta_C \frac{\eta_T \frac{T_O}{T_{AC}}\left(1 - Er^{\frac{1-k}{k}}\right) - \frac{T_O - T_{CI}}{T_{AC}}}{\left(Er^{\frac{k-1}{k}} - \eta_C \eta_T \eta_M \frac{T_O}{T_{CI}}\right)\left(Er^{\frac{k-1}{k}} - 1\right)}\left[\eta_C \eta_T \eta_M \frac{T_O}{T_{CI}}\left(Er^{\frac{k-1}{k}} - 1\right) + Er^{\frac{k-1}{k}}\right] \tag{10}$$

$$\text{COP}_{\text{C}-\text{TC cycle}} = \eta_C \frac{\eta_T \frac{T_O}{T_{CI}}\left(1 - Er^{\frac{1-k}{k}}\right) - \frac{T_O - T_{CI}}{T_{CI}}}{\left(Er^{\frac{k-1}{k}} - \eta_C \eta_T \eta_M \frac{T_O}{T_{AC}}\right)\left(Er^{\frac{k-1}{k}} - 1\right)}\left[\eta_C \eta_T \eta_M \frac{T_O}{T_{AC}}\left(Er^{\frac{k-1}{k}} - 1\right) + Er^{\frac{k-1}{k}}\right] \qquad (11)$$

The COP of the two open-loop high-pressure cycles (C-CT cycle and CT-C cycle) can be determined by Equations (12) and (13), respectively:

$$\text{COP}_{\text{C}-\text{CT cycle}} = \eta_C \frac{\eta_T \frac{T_{AC}}{T_I}\left(1 - Er^{\frac{1-k}{k}}\right) - \frac{T_{AC} - T_I}{T_I}}{\left(Er^{\frac{k-1}{k}} - \eta_C \eta_T \eta_M\right)\left(Er^{\frac{k-1}{k}} - 1\right)}\left[\eta_C \eta_T \eta_M \left(Er^{\frac{k-1}{k}} - 1\right) + Er^{\frac{k-1}{k}}\right] \qquad (12)$$

$$\text{COP}_{\text{CT}-\text{C cycle}} = \eta_C \frac{\eta_T \left(1 - Er^{\frac{1-k}{k}}\right) - \frac{T_{AC} - T_I}{T_{AC}}}{\left(Er^{\frac{k-1}{k}} - \eta_C \eta_T \eta_M \frac{T_{AC}}{T_I}\right)\left(Er^{\frac{k-1}{k}} - 1\right)}\left[\eta_C \eta_T \eta_M \left(Er^{\frac{k-1}{k}} - 1\right) + Er^{\frac{k-1}{k}}\right] \qquad (13)$$

In the above case, where $T_I$ = 300.15 K, $T_O$ = 308.15 K, $T_{AC}$ = 313.15 K, $T_{CI}$ = 295.15 K, $\eta_T$ = 0.8, $\eta_C$ = 0.75, $\eta_M$ = 0.99, variations of COP with expansion ratio for various cycles are shown in Figure 3. It can be observed that the COP of the closed cycle is lower than that of the open-loop cycles due to the loss caused by the finite temperature difference in the heat exchanger. The four two-stage open-loop cycles have higher COPs, especially when the expansion ratio is larger than 1.7.

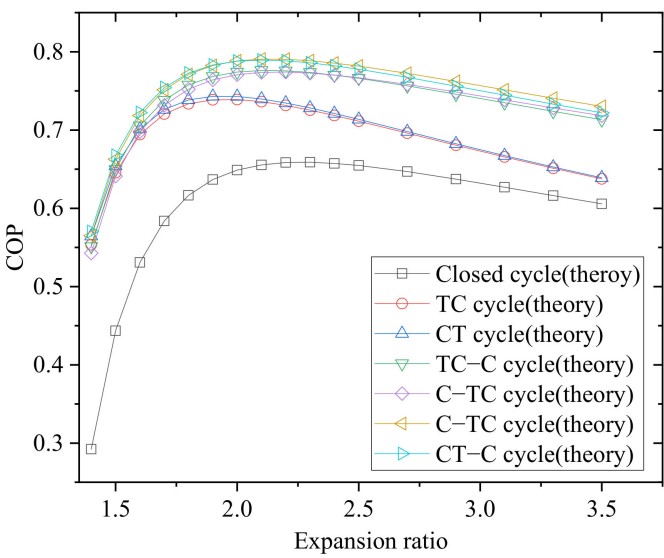

**Figure 3.** Variations in COP as a function of expansion ratio for various air refrigeration cycles.

However, in a practical cycle, the working medium in the above open-loop cycles is wet air, and the energy loss caused by the condensed water cannot be ignored. When the expansion ratio is relatively large, the air temperature after the compressor and the expander may be lower than the dew point, and the water vapor is condensed. In order to control the humidity, the air containing condensed water cannot be directly supplied to the room. As the condensed water is removed, the mass flow rate is reduced, resulting in the loss of cooling capacity. Therefore, the cycle performance in a practical air refrigerator depends on the humidity and pressure of the ambient air, in addition to the losses in expanders and compressors, which are discussed in the following section. In order to consider the effect of air temperature and absolute humidity, the enthalpy of wet air can be calculated by Equation (14):

$$h_{\text{wet}} = c_{p,a}T + d\left(c_{p,v}T + L\right) \qquad (14)$$

where $c_{p,a}$ is the specific heat at constant pressure of dry air, which is mostly related to temperature, and $c_{p,v}$ is the specific heat at constant pressure of water vapor. *L* is the latent

heat of the water vapor. The above data can be obtained from the NIST REFPROP database to fit the corresponding correlations; $d$ is the absolute humidity of the wet air.

The following part takes the TC cycle as an example to illustrate the thermodynamic analysis on the actual system performance of the open-loop cycle. Although the internal expansion process in the turbo-expander is too fast for the water to condense, the moist air in the downstream pipeline gradually reaches the equilibrium state as the water vapor is condensed. Therefore, the latent heat of vapor condensation needs to be considered. In the calculation, and it is assumed that no water is condensed inside the expander, and the influence of water on the isentropic efficiency of the expander is ignored. The calculation process is shown in Figure 4. In the process, the outlet temperature of the expander $T_{EO}$ and the supplied temperature $T_{sup}$ are solved iteratively, according to the change in enthalpy.

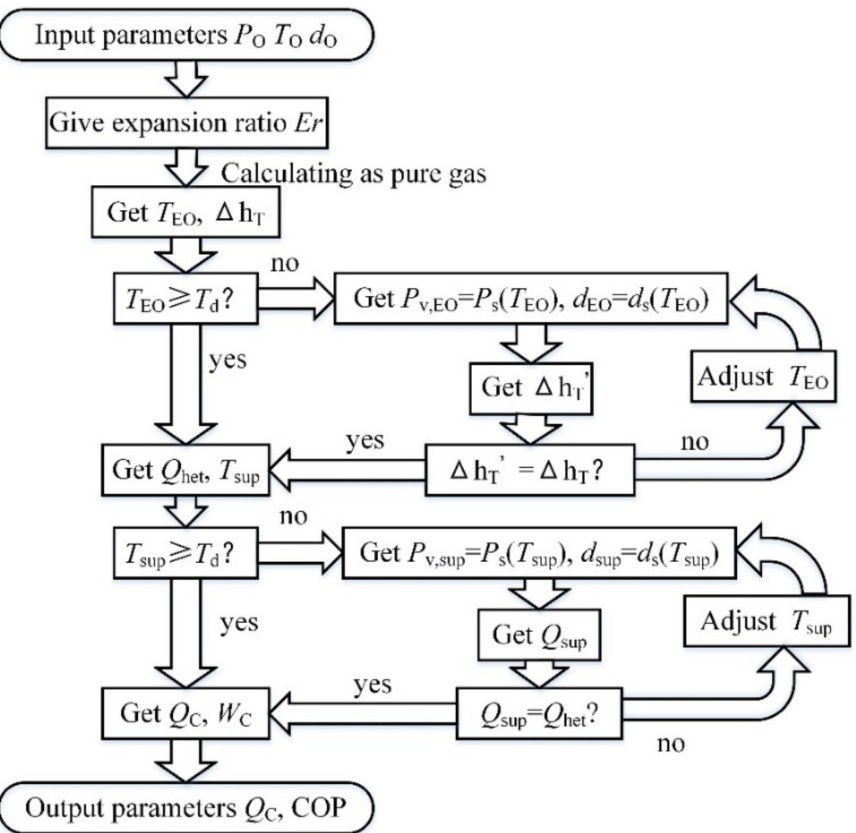

**Figure 4.** Calculation process of thermodynamic performance for a practical TC cycle.

## 3. Results and Discussion

In an air refrigeration cycle, a counter-current flow heat-exchanger is commonly utilized as a recuperator. Considering the limitations of pressure loss and size, it is assumed that the minimum temperature difference in the heat exchanger is 5 K. The design parameters for the thermodynamic analysis of air refrigeration cycles are shown in Table 1. The working fluid in the closed cycle is dry air to avoid the energy loss due to condensation of water vapor in the moist air. The open-loop cycle intakes air from the ambient environment, which contains water vapor. When the outlet temperature of the expander is lower than the dew point, the water vapor is condensed, which has an impact on the outlet temperature of the expander. The influence of the latent heat of water condensation is considered in the calculation of the outlet temperature of the expander.

**Table 1.** Ambient conditions and design parameters of air refrigeration cycles.

| Work Fluid | Ambient Pressure | Dry Bubble Temperature (Indoor) | Wet Bubble Temperature (Indoor) |
|---|---|---|---|
| Moist Air | 101.325 kPa | 300.15 K | 292.15 K |
| **Dry Bubble Temperature (Outdoor)** | **Wet Bubble Temperature (Outdoor)** | **Isentropic Efficiency (Compressor)** | **Isentropic Efficiency (Expander)** |
| 308.15 K | 297.15 K | 75% | 80% |
| **Pressure Drop (Water Separator)** | **Pressure Drop (Heat Exchanger)** | **Mechanical Efficiency** | **Temperature Difference (Heat Exchanger)** |
| 1.0 kPa | 1.0 kPa | 99% | ≥5 K |

For closed air refrigeration cycles with varied expander inlet pressure, Figure 5 shows the variations in COP and supplied-air temperature with the expansion ratio. It is apparent that the increase in operating pressure can improve cycle COP, but it has little effect on the supplied-air's temperature. As operating pressure increases, the enhancement of the cycle COP becomes less obvious. This is mainly because the system power consumption and cooling capacity mainly depend on the enthalpy difference when the pressure is beyond 0.3 MPa. Figure 5 shows that there is an optimal expansion ratio to maximize the cycle COP. When the inlet pressure of the expander is 506.625 kPa, the maximum COP of 0.652 can be reached at an expansion ratio of 2.25.

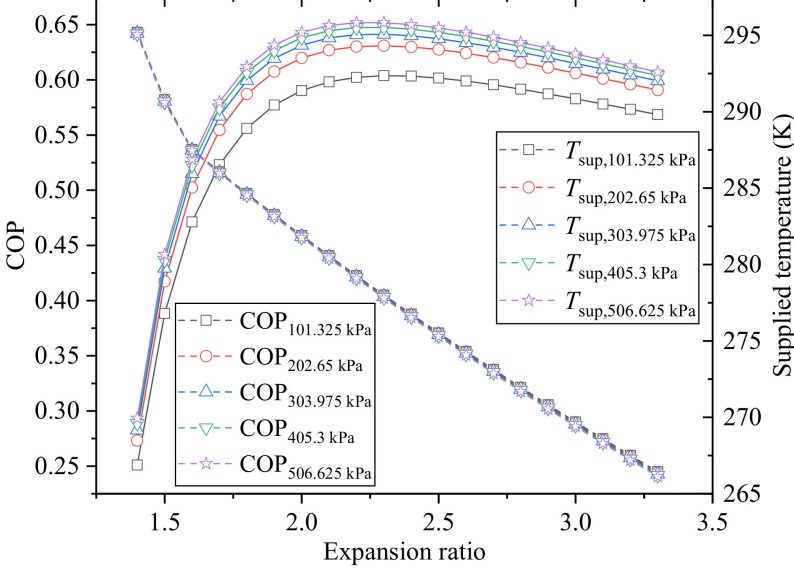

**Figure 5.** Comparisons of COP and supplied-air temperature for closed air-refrigeration cycles under different expander inlet pressures.

Figure 6a shows the comparisons of the COP and supplied-air temperature for various air refrigeration cycles. The COP of open-loop cycles is greater than closed-loop cycles, and the two-stage compression open-loop cycles, including TC-C, C-TC, C-CT and CT-C cycles have higher COPs than the one-stage compression open-loop cycles. The greatest COP for the TC cycle and CT cycle are 0.728 and 0.747, respectively, for the optimal expansion ratios of 2.10 and 2.17. Figure 6b shows the p-H diagrams for the air cycles at the respective optimal expansion ratio. In an open-loop high-pressure cycle (CT cycle), the cold air from the expander can be supplied to the compartment directly, and there is no loss of cooling capacity caused by the temperature difference in the cold-side heat exchanger. The inlet temperature of the expander in an open-loop low-pressure cycle (TC cycle) is the ambient temperature, which is generally lower than that of the CT cycle, and the cooling capacity

can be increased effectively. However, the temperature difference in the cold-side heat exchanger of the TC cycle leads to a lower COP than that of the CT cycle, and the influence is more obvious when the expansion ratio is higher than the optimum value.

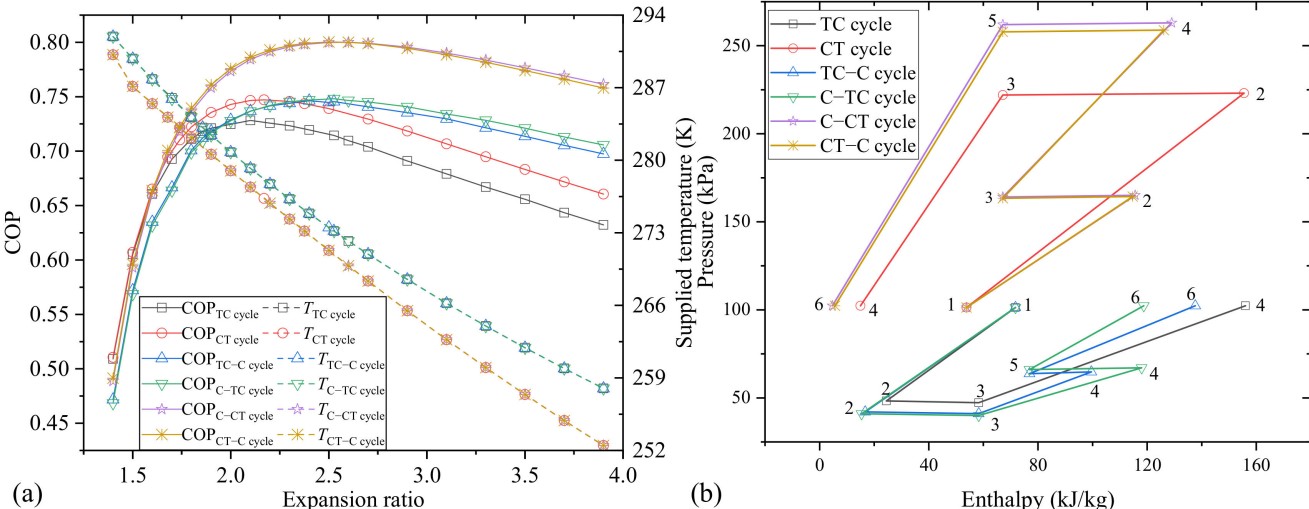

(a)

(b)

**Figure 6.** (**a**) Comparisons of COP and supplied-air temperature for different air refrigeration cycles; (**b**) p-H diagrams for each cycle at the optimal expansion ratio.

The optimal expansion ratios of the TC-C cycle, C-TC cycle, C-CT cycle and CT-C cycle are 2.41, 2.47, 2.56 and 2.52, respectively, and the corresponding maximum COPs are 0.746, 0.748, 0.800 and 0.800, respectively. When the expansion ratio is less than 1.9 for open-loop low-pressure cycles, the COP of the TC cycle is higher than that of the TC-C cycle and C-TC cycle. As the expansion ratio increases, the COP of the C-TC cycle and TC-C cycle surpasses the TC cycle, and the COP of the TC-C cycle is smaller than that of the C-TC cycle. When the supplied-air temperature is higher than about 282 K, the COP of the TC cycle is highest; otherwise, the COP of the C-TC cycle is highest. Similarly, the COP of the CT cycle is higher than that of the C-CT cycle and CT-C cycle when the expansion ratio is less than 1.6. As the expansion ratio increases, the COP of the C-CT cycle and CT-C cycle gradually exceeds the TC cycle. The CT cycle has the highest COP when the supplied-air temperature is higher than 285 K. The COP of the C-CT cycle is slightly higher than that of the CT-C under other conditions.

In addition, air refrigeration systems such as domestic air conditioners also have requirements for the supplied temperature. The supplied temperature is generally required to be greater than 283.15 K to ensure the comfort of human body, which requires a relatively low expansion ratio. This makes the open-loop cycle unable to operate at the optimal expansion ratio. At this case, the performance of CT-C cycle is the best (COP is 0.722). The COP of TC cycle is somewhat lower than that of CT-C cycle, but it has a simpler configuration with one-stage compression.

The expansion ratio should not be too large to prevent ice in the system when the supplied-air temperature is lower than 273.16 K. Open-loop low-pressure cycles can achieve the optimal pressure ratio, but their COP is lower than that of open-loop high-pressure cycles. The C-CT cycle and CT-C cycle have the highest COP of 0.798 when the supplied-air temperature is above 273.16 K. Alternatively, before being supplied to the compartment, the cold air from expander can be mixed with the indoor warm air. The temperature at expander outlet can be lower than 273.16 K in this situation, and the cooling capacity can be increased by increasing the expansion ratio.

### 3.1. The Effect of Expander and Compressor Efficiencies

Figure 7 reveals variations in COP and supplied-air temperature with the expansion ratio for open-loop high-pressure and open-loop low-pressure cycles. It is apparent that the

COP of the open-loop high-pressure cycles is larger than that of the open-loop low-pressure cycles, where the COP of the CT-C is the largest when the expansion ratio is greater than 1.8. When the expansion ratio is greater than 1.9 or the supplied-air temperature is lower than 281 K, the COP of the C-CT is the largest in the open low-pressure cycles. As the efficiencies of the compressor and expander increase, the COP of the open-loop cycles increases a lot, and the optimal expansion ratio decreases. The optimal expansion ratios of the TC cycle (85% and 80%), TC-C cycle (85% and 80%) and C-TC cycle (85% and 80%) are 1.96, 2.25 and 2.31, respectively, with maximum COP values of 0.974, 0.997 and 0.979, respectively.

In the open high-pressure cycles, when the expansion ratio is greater than 1.6 or the supplied temperature is lower than 286 K, the COP of the CT-C cycle is the largest. The optimal expansion ratios of the CT cycle (85% and 80%), C-CT cycle (85% and 80%) and CT-C cycle (85% and 80%) are 2.04, 2.41 and 2.35, respectively, and the corresponding maximum COPs are 0.997, 1.055 and 1.071, respectively. When the supplied-air temperature required is greater than 283.15 K, the CT-C cycle has the maximum COP of 0.969, while the TC cycle has almost the same COP (0.967) but a simpler system arrangement. When the supplied-air temperature is greater than 273.16 K, the CT-C cycle has the maximum COP of 1.069.

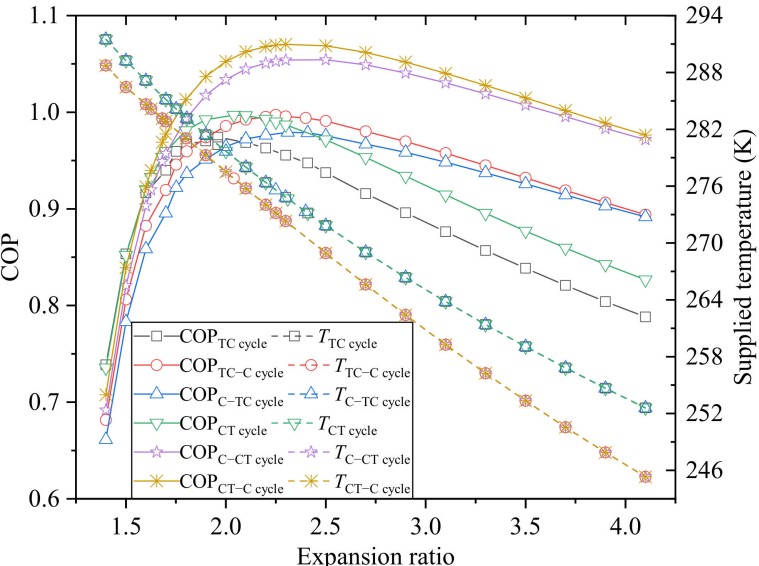

**Figure 7.** Comparison of COP and supplied-air temperature when the isentropic efficiency of expander and compressor is 85% and 80%, respectively.

### 3.2. The Effect of Humidity

Figure 8a shows variations in COP with the expansion ratio for the CT cycles with different indoor wet bulb temperatures. Figure 8b shows the p-H diagrams for the air cycles at the respective initial optimal COP. The isentropic efficiencies of the expander and compressor are 80% and 75%, respectively. In the legend, for example, 300.15 K/292.15 K denotes the dry/wet bulb temperatures. When the indoor humidity is high, there is a turning point in the curve of the COP that varies with the expansion ratio. In the zone after the turning point, the COP increases rapidly with the increase of the expansion ratio, and the temperature of after-cooler outlet is lower than the dew point. The latent heat of phase change allows the working fluid to absorb more cooling capacity in the after-cooler, which results in a decrease of absolute moisture in the air and a lower enthalpy at the expander inlet. Although the mass flow decreases slightly, the decrease in enthalpy has a greater impact on the cooling capacity of the system. Therefore, when the outlet temperature of the after-cooler is lower than the dew point, the cooling capacity of the system is greater. In addition, with the humidity increases, both the COP and the optimal expansion ratio of the cycle increase. Before the turning point, the optimal expansion ratios are 2.17, 2.20, 2.20,

2.22 and 2.24, respectively, and the corresponding maximum COPs are 0.747, 0.756, 0.766, 0.776 and 0.785, respectively.

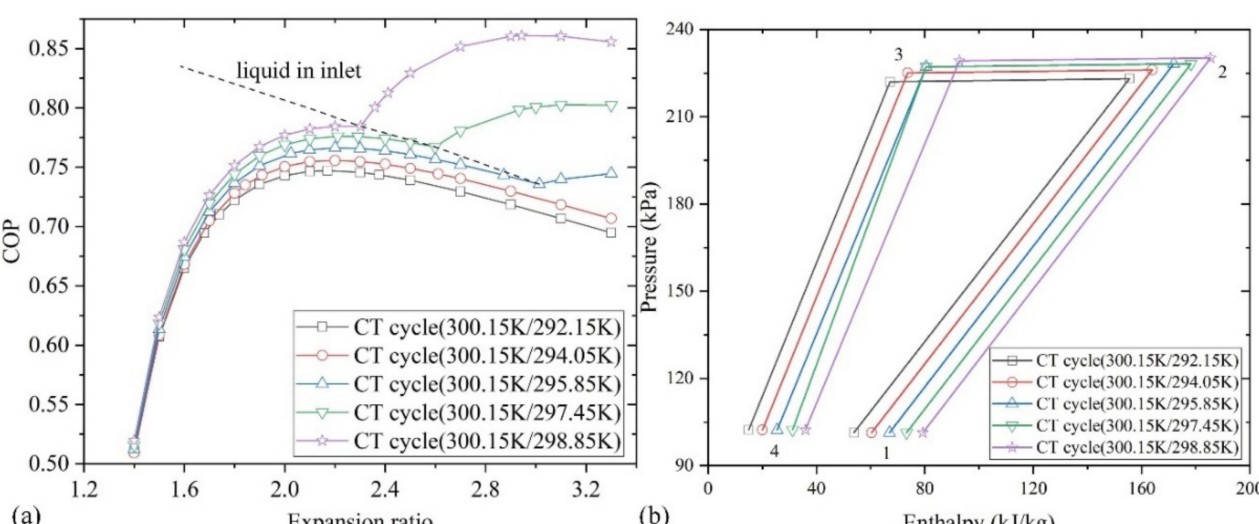

**Figure 8.** (**a**) Comparison of COP of open-loop high pressure cycle under different indoor wet bulb temperature conditions; (**b**) p-H diagrams for each cycle at the initial optimal COP.

### 3.3. The Effect of Recycling Water

The low-temperature water is directly discharged at the outlet of the air refrigeration cycle, and the cooling capacity is also wasted. If the cooling capacity of the water that separated from the separator can be recycled, the COP of the system will improve. For an open-loop low-pressure cycle, the water separated from the supplied side can be mixed with the air at the expander inlet without bringing additional wet load to the supplied side, which can reduce the temperature of the expander inlet. The water can be atomized and injected as much as possible into the inlet of the expander to cool the humid air of the expander inlet to a saturated state. In order to avoid problems such as the corrosion of parts caused by liquid at the turbo-expander inlet, the mass flow of the recirculating water should not be so large that the moist air of the expander inlet is supersaturated. The power consumption of the atomized water is ignored in the calculations. When the mass flow of the condensed water is greater than what the expander inlet requires, the remaining water can be atomized and injected into the compressor inlet, which makes a wet compression in the compressor. During the compression, the water evaporates and absorbs the compression heat, reducing the power consumption. The reduced theoretical compression work is equal to the latent heat of the water, and the effect of the increased mass flow is also calculated.

Figure 9 shows the variations in COP with the expansion ratio in open-loop low-pressure cycles with recycling water. Without wet compression, the optimal expansion ratio occurs when the mass flow of the recycling water just makes the wet air of the expander inlet reach a saturated state. When the expansion ratio is larger than the optimal expansion ratio, the temperature of the expander inlet cannot continue to decrease, reducing the effect of the recycling water on the cooling capacity of the system. The maximum COP of the TC, TC-C and C-TC cycles are 1.310, 1.297 and 1.268, respectively. Compared to the COP before recycling the water, the maximum COP of the TC, TC-C and C-TC cycles have increased by 34.5%, 30.1% and 29.5%, respectively. The optimal expansion ratio of the TC, TC-C and C-TC cycles is 1.78. At the same time, the supplied-air temperature of the system is higher than 283.15 K. When the remaining water is used for wet compression, the optimal expansion ratios of the TC, TC-C and C-TC cycles are 2.12, 2.32 and 2.35, respectively, and the corresponding maximum COP are 1.410, 1.483 and 1.453, respectively. Compared to

the COPs before recycling the water, the maximum COPs of the TC, TC-C and C-TC cycles have increased by 44.7%, 48.8% and 48.4%, respectively.

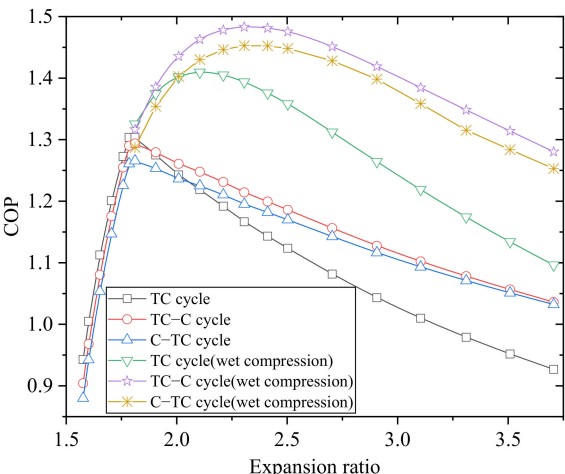

**Figure 9.** COP of air refrigerators with recycling water with isentropic efficiency of expander and compressor at 85% and 80%, respectively.

### 3.4. The Impact of Fresh Air

A fresh-air system can significantly improve the indoor air quality in a closed indoor environment. Simultaneously, in order to inhibit the spread of bacteria and viruses, it is also necessary to use an all-fresh-air system without return air. However, the introduction of fresh air brings additional heat and wet load, thereby reducing the cooling capacity of the system. The COP of open-air cycles with the all-fresh-air system is studied.

Figure 10 shows the schematic of the open-loop air refrigeration cycles with all-fresh-air supply. The exhaust side of an open-loop low-pressure cycle intakes air indoors, while the supplied side sucks air outdoors and feeds the conditioned air indoors. The supplied side of an open-loop high-pressure cycle intakes outdoor air and feeds the conditioned air indoors, and the indoor exhaust air cools the air of the expander inlet.

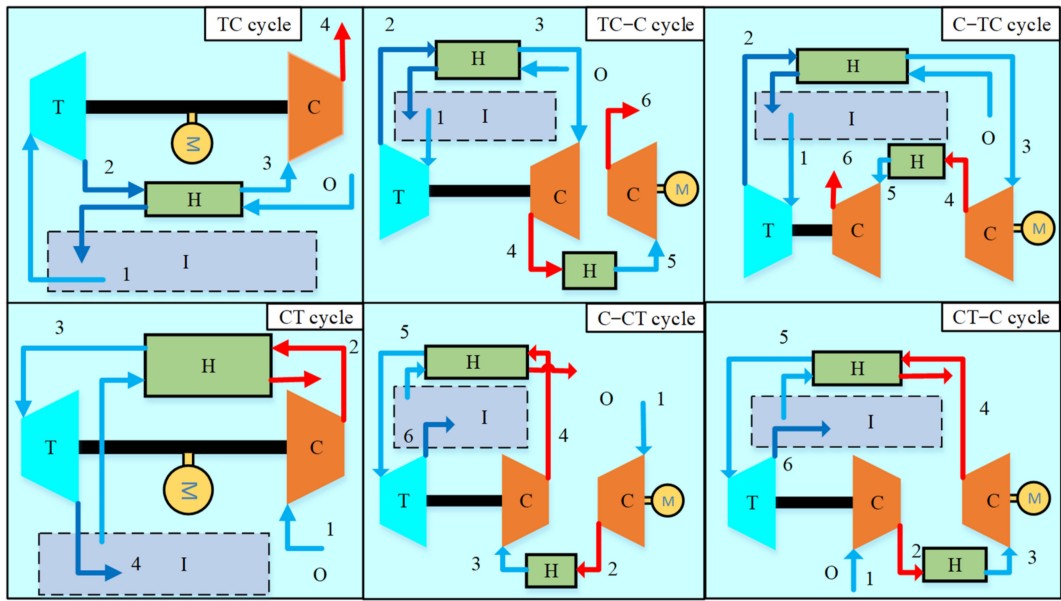

**Figure 10.** Schematic of the all-fresh-air system for open-loop air refrigeration cycles. T: turbo expander; C: compressor; I: indoor; O: outdoor.

Figure 11a reveals the variations in COP and supplied-air temperature with the expansion ratio for air refrigeration cycles with all-fresh-air systems when the isentropic efficiency of the expander and compressor is 80% and 75%, respectively. Figure 11b shows the p-H diagrams for the air cycles at the respective optimal expansion ratios. The optimal expansion ratios of the TC cycle, TC-C cycle, C-TC cycle, CT cycle, C-CT cycle and CT-C cycle are 2.25, 2.56, 2.59, 2.37, 2.76 and 2.74, respectively, and the corresponding maximum COPs are 0.620, 0.655, 0.659, 0.661, 0.739 and 0.738, respectively. Comprehensive comparison shows that when the required supplied-air temperature is greater than 283.15 K, the CT-C cycle has the best COP (0.646), which is close to that of the C-CT cycle (0.645). When the required supplied-air temperature is greater than 273.16 K, the C-CT cycle has the best COP (0.734). Adopting the all-fresh-air system, the optimal expansion ratio increases. In open-loop low-pressure cycles, the C-TC cycle has the best COP.

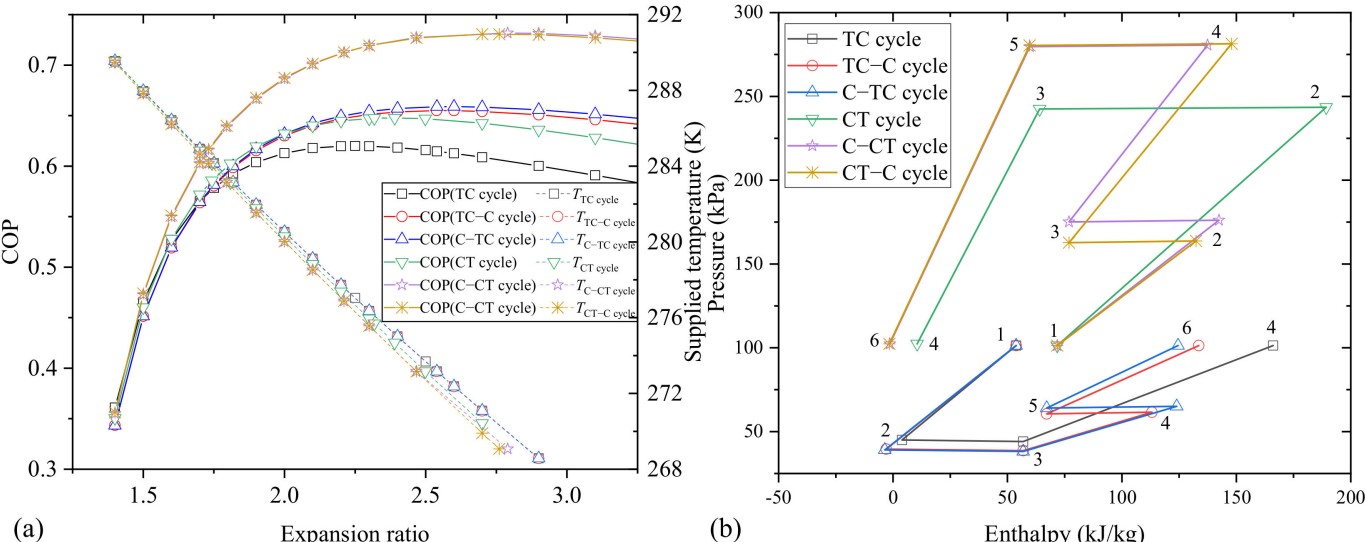

**Figure 11.** (**a**) COP and supplied-air temperature of air refrigerators with all-fresh-air systems with isentropic efficiency of expander and compressor of 80 and 75%, respectively; (**b**) p-H diagrams for each cycle at the optimal expansion ratio.

Figure 12 illustrates the variations in COP with the expansion ratio and supplied temperature for open-loop air refrigeration cycles with all-fresh-air systems when the expander and compressor isentropic efficiency is 85% and 80%. The difference from the previous example is that the COP of the TC-C cycle is greater than that of the C-TC cycle. The CT-C cycle has the best COP of the high-pressure cycles. The optimal expansion ratios of the TC cycle, TC-C cycle, C-TC cycle, CT cycle, C-CT cycle and CT-C cycle are 2.09, 2.39, 2.41, 2.23, 2.60 and 2.58, respectively, and the corresponding maximum COPs are 0.816, 0.863, 0.858, 0.872, 0.972 and 0.979, respectively. Similarly, when the supplied-air temperature is required to be greater than 283.15 K or 273.16 K, the CT-C cycle has the best COP. The TC-C cycle has the best COP in the open-loop low-pressure cycles.

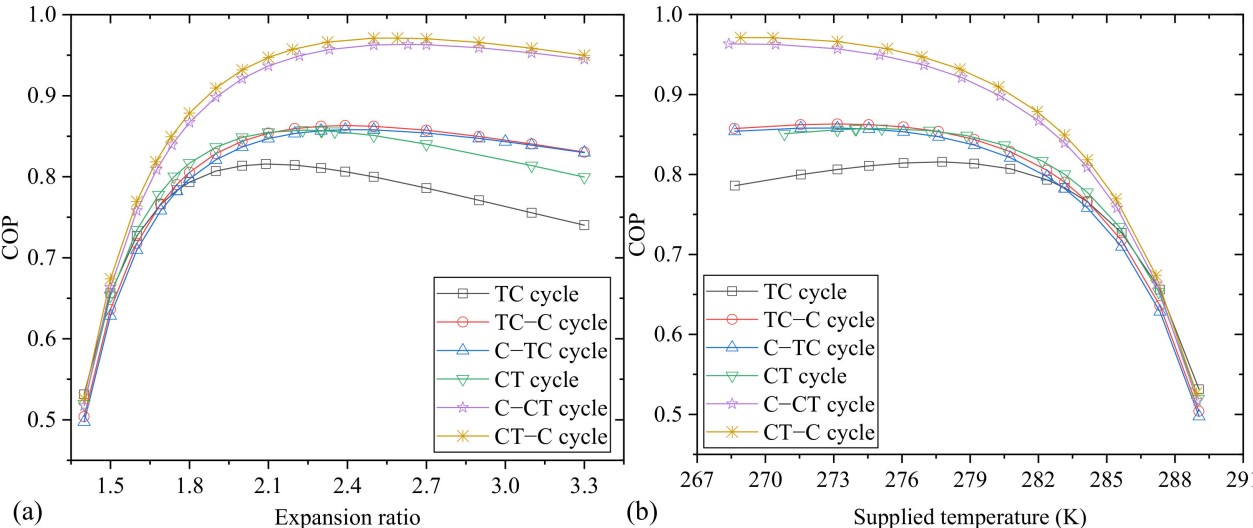

**Figure 12.** COP of air refrigerators with all-fresh-air systems with isentropic efficiency of the expander and compressor of 85% and 80%, respectively: (**a**) variable expansion ratio and (**b**) variable supplied temperature.

*3.5. The Effect of Ambient Pressure of High-Altitude Regions*

When trains are running from the plain area to the plateau area, air conditioning with elevated supply pressure and fresh air is desired in order to maintain a comfortable oxygen concentration. On the plateau train of the Qinghai–Tibet line in China, there are more than 960 km of railways where the altitude is above 4000 m and the atmospheric pressure is below 56.04 kPa, and the highest altitude is 5072 m. In that case, the respiratory and digestive systems of the human body are significantly affected and uncomfortable. As required by passenger aircraft, cabin pressure should be generally maintained around 72.84–80.9 kPa. For the conventional vapor compression air-conditioning systems, a separate fresh-air system with an air compressor is needed. Considering the sensible heat and compression work of the fresh air, the COP of the vapor compression air-conditioning system decreases significantly to a level which is comparable to that of the air-cycle refrigeration system.

For the application of an open-loop low-pressure cycle in a high-altitude train, it is necessary to employ a compressor on the supplied side to increase the pressure of outdoor fresh air. In an open-loop high-pressure cycle, an expander can be used on the cabin exhaust side to recover the pressure energy. It is assumed that the isentropic efficiencies of the expander and compressor are 85% and 80%, respectively, and the supplied air pressure is 72.84 kPa. Indoor and outdoor temperatures remain same as in previous calculations, and the absolute humidity of the indoor and outdoor air remain unchanged.

Figure 13 shows the variations in COP with the expansion ratio and supplied-air temperature for open-loop cycles when the ambient pressure is 56.04 kPa. The COP of the open-loop low-pressure cycle is better. Optimal expansion ratios of the TC, TC-C, C-TC, CT, C-CT and CT-C cycles are 2.43, 2.85, 2.93, 2.79, 3.40 and 3.40, respectively, and the corresponding maximum COP are 0.766, 0.776, 0.761, 0.688, 0.782 and 0.789, respectively. When the supplied-air temperature is greater than 283.15 K or 273.16 K, the TC cycle has the maximum COP of 0.588 or 0.763. It is obvious that the COP decreases after the atmospheric pressure is lowered. Under the same absolute humidity, the partial pressure of water vapor is lower than that under normal pressure, and the dew point of the air is lower. Under the same enthalpy drop, the temperature drop of the moist air is greater, and thus the temperature of the expander is lower. In order to increase the supplied-air temperature, it is necessary to mix the air from the expander with indoor air before supplying the air to the cabin. It is not until the supplied temperature is very low that the COP of the CT-C cycle and CT-C cycle are better. However, there is high-pressure air in the high-pressure cycles that can be used to generate high-concentration oxygen.

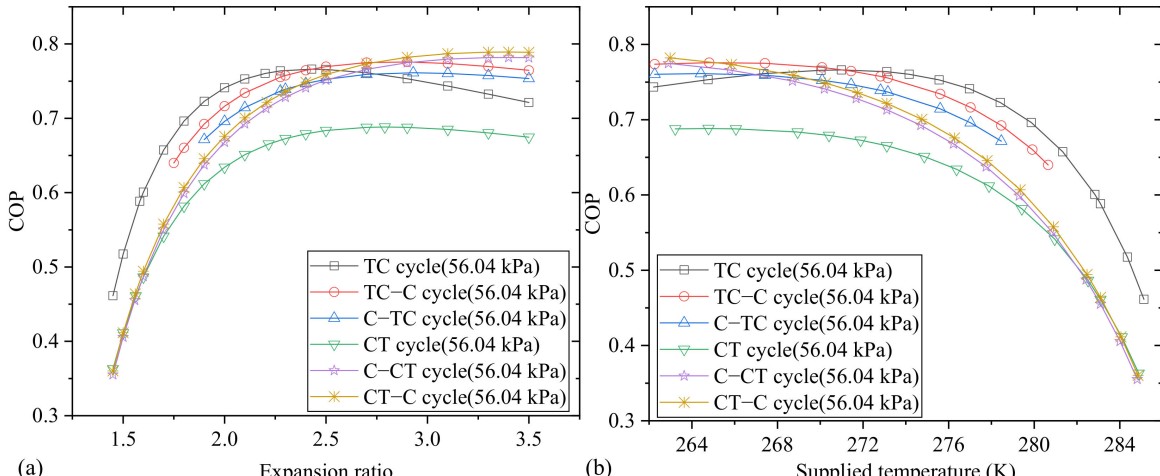

**Figure 13.** COP of air refrigerators in high altitude trains at the ambient pressure of 56.04 kPa for all fresh-air systems: (**a**) variable expansion ratio and (**b**) variable supplied temperature.

For the air-conditioning systems of plateau trains, the outside pressure also changes during the trip, especially when the train travels from the plain to the plateau. The air refrigeration cycle with the centrifugal compressors and turbo-expanders can adjust the system pressure ratio by changing the rotational speed to match the ambient pressure. Although the compression power is required to maintain the indoor pressure, the air-cycle refrigeration system can use the expander to recover the pressure energy of the indoor exhaust air, and therefore the cycle COP does not decrease too much compared to that of the plain.

## 4. Conclusions

In this study, we investigated the thermodynamic performance of three configurations of the air cycle refrigeration system for compartment air conditioning, including closed-loop cycles, open-loop low-pressure cycles and open-loop high-pressure cycles. The results show that the COP of the closed-loop cycle is generally smaller than that of the open-loop cycles because of the irreversible loss in the heat exchangers. The following conclusions can be drawn.

(1) For all the air refrigeration cycles, the optimal expansion ratio decreases with increase of the compressor and expander efficiency, but it increases with the decrease of the ambient pressure. For all-fresh-air air-conditioning, the optimal expansion ratio is increased, while the corresponding COP is reduced. Both the optimal expansion ratio and COP are increased in the two-stage compression intercooling configuration.

(2) For the normal environment, among open-loop cycles, high-pressure cycles have higher COP than low-pressure cycles. In open-loop low-pressure cycles, the COP of the TC-C cycle exceeds that of the CT-C cycle as the compressor efficiency increases, and a similar trend can also be observed in open-loop high-pressure cycles.

(3) An increase in air humidity increases the optimal expansion ratio and COP of the system. In open-loop low-pressure cycles, the condensed water can be recycled by being sprayed into the expander inlet, increasing the maximum COP of the TC, TC-C and C-TC cycles by 34.5%, 30.1% and 29.5%, respectively. Furthermore, if the remaining water is used for wet compression, the maximum COP of the TC, TC-C and C-TC cycles can be increased by 44.7%, 48.8% and 48.4%, respectively.

(4) In the air conditioning systems of plateau trains, the open-loop low-pressure cycle has the highest COP under the requirement that the supplied temperature is greater than 283.15 K. The COPs of open-loop high-pressure cycles are relatively low, but the high-pressure air in the cycle can be used to separate high-concentration oxygen.

The COP of the air refrigeration cycle at room temperature is low, and humidity can improve the COP. Future research needs to further improve the efficiency of the

compressor and expander to improve the COP. Improving the efficiency of wet expansion and compression, in particular, can improve the COP.

**Author Contributions:** Conceptualization, X.Y. and L.C.; data curation, X.Y.; formal analysis, Z.Z.; investigation, X.Y.; methodology, X.Y. and Y.H.; project administration, S.C.; resources, L.C.; validation, X.Y. and Z.W.; visualization, X.Y.; writing—original draft preparation, X.Y.; writing—review and editing, X.Y. and L.C. All authors have read and agreed to the published version of the manuscript.

**Funding:** This project was supported by the National Nature Science Foundation of China (U21B2084) and the Youth Innovation Team of Shaanxi Universities.

**Data Availability Statement:** Not applicable.

**Conflicts of Interest:** The authors declare no conflict of interest.

## Nomenclature

| | |
|---|---|
| COP | coefficient of performance |
| $c_p$ | specific heat at constant pressure (J kg$^{-1}$) |
| d | humidity ratio of wet air (g kg$_a^{-1}$) |
| $Er$ | expansion ratio |
| $h$ | enthalpy (J kg$^{-1}$) |
| $k$ | adiabatic exponent |
| $P$ | pressure (Pa) |
| $Q$c | cooling capacity (W) |
| $Q_{het}$ | exchanger cold duty (W) |
| $q_m$ | mass flow rate (kg s$^{-1}$) |
| $R$ | gas constant of air (J kg$^{-1}$K$^{-1}$) |
| $T$ | temperature (K) |
| $T_d$ | dew point temperature (K) |
| W | shaft power (W) |
| Greek | |
| $\eta$ | isentropic efficiency |
| $\eta_M$ | mechanical efficiency |
| $\Delta$h | the change of enthalpy (J kg$^{-1}$) |
| Subscripts | |
| a | dry air in wet air |
| AC | outlet of the after-cooler |
| C | compressor |
| CI | cold-side heat exchanger outlet |
| EO | turbo-expander outlet |
| I | indoor |
| max | maximum |
| O | outdoor |
| opt | optimal |
| s | saturated |
| sup | supplied air |
| T | turbo-expander |
| v | water vapor in wet air |

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
