# Peer review of "Thermodynamic Analysis of Air-Cycle Refrigeration Systems with Expansion Work Recovery for Compartment Air Conditioning"

_applsci, doi:10.3390/app12105287_

Round 1

Reviewer 1 Report

The manuscript presented the thermodynamic performance of an air cycle refrigeration system with three types of configurations. In particular, authors introduced an expander to maintain the COP of the refrigeration cycle. The following comments have to be accommodated.

  1. Authors could include some quantitative results (optimum COP, humidity etc.) in the abstract.
  2. Despite of adding the findings of previous studies; authors could include the drawbacks of these literatures and how the present study overcame it.
  3. Highlight the novelty of the study in the last paragraph of the introduction section (only research gap is mentioned).
  4. Check the first line of the 2nd section first paragraph.
  5. Wherever possible, support the arguments or phenomenon with previous literatures (For example, “by integrating the expander with one of the compressors, the matching characteristics of the aerodynamic performance can be improved while the expansion work is recovered”)
  6. Figure 3 seems to be one of the results, it should be placed in results section.
  7. Please improve the resolution/quality of the figures. Figure should be placed uniformly all throughout the manuscript (especially Figure 12 & 13).
  8. If possible, validate or compare the obtained results with previous study to ensure its credibility.
  9. Authors could include some drawbacks or future recommendations or applications to pave a way for a new study in the conclusion section.
  10. Include references from recent studies also.

Reviewer 2 Report

The introduction is to be extended by adding some new recent published papers.
The English level is to be improved.
Are the results obtained by direct calculations or iterative calculations? If it is iterative, what is the used method?
Higher resolution is to be provided for the figures.
A nomenclature is to be added.
In figure 8, what is the relation between the liquid inlet and the expansion ratio?

Reviewer 3 Report

The paper is to be checked against mistakes and grammatical mistakes.

Some figures are not Sharpe enough.

What is the novelty of the present work? The calculations are very simple.

The authors must add a nomenclature.

Why are the results limited to COP? The cycles are to be plotted in p-H and/or T-s diagrams.

 The design parameters presented in Table 1 are to be justified.

In the calculations, is Er given or calculated? If it is given, it is to be considered as input parameter.

What do you mean by ‘’do’’ in the input parameters (Fig 4)

In figure 7, why Er is above for 4 the variations of COP but is limited to 3.25 and 3.75 for the variations of supplied temperature.

Round 2

Reviewer 3 Report

After revision the paper can be accepted for publication